# Socio-Demographic Factors Linked to Psychological Well-Being in Dementia Caregivers

**DOI:** 10.3390/healthcare13172235

**Published:** 2025-09-07

**Authors:** Liviu Florian Tatomirescu, Cristiana Susana Glavce, Gabriel Ioan Prada, Adriana Borosanu, Suzana Turcu

**Affiliations:** 1Francisc I. Rainer Institute of Anthropology, Romanian Academy, 050474 Bucharest, Romania; liviufloriant@yahoo.com (L.F.T.); glavcecristiana@yahoo.fr (C.S.G.); adriana_borosanu@yahoo.com (A.B.); 2“C.F.2” Clinical Hospital, 011464 Bucharest, Romania; 3National Institute of Gerontology and Geriatrics “Ana Aslan”, 011241 Bucharest, Romania; giprada@gmail.com

**Keywords:** psychological well-being, caregivers, dementia, depression, anxiety

## Abstract

**Background:** Caregivers of individuals with cognitive impairment face heightened emotional and psychological burdens, yet the interaction between caregiver well-being, patient characteristics, and socio-demographic factors still requires investigation. This study aimed to examine the psychological well-being of family caregivers in an urban Romanian context, focusing on the role of depressive and anxiety symptoms, education, and care-recipient cognition function. **Methods:** A cross-sectional study was conducted among family caregivers recruited from a neurology-psychiatry service in Bucharest. Caregivers completed Ryff’s Psychological Well-Being Scales, the Patient Health Questionnaire-9 (PHQ-9), and the COVI Scale. Cognitive status of care recipients was obtained from medical records (Mini-Mental State Examination, MMSE). Descriptive statistics, correlation analyses, and separate linear regression models were performed for each well-being dimension. **Results:** Caregivers reported moderate to high well-being scores, with Environmental Mastery highest (M = 38.01, SD = 8.70) and Purpose in Life lowest (M = 33.14, SD = 6.72). Depression scores averaged 18.49 (SD = 6.55), indicating moderate depressive symptoms, and anxiety scores averaged 12.14 (SD = 2.23), consistent with severe anxiety. Cognitive impairment in care recipients was marked (MMSE M = 11.47, SD = 6.99). Bivariate analyses showed that lower MMSE scores were associated with higher caregiver anxiety (ρ = −0.287, *p* = 0.014). Regression models (R^2^ = 0.08–0.25) indicated that higher education was positively associated with autonomy, personal growth, positive relations, and environmental mastery, whereas older age and female gender were linked to lower well-being in several domains. Depressive symptoms were unexpectedly associated with higher autonomy and self-acceptance. **Conclusions:** Caregiver psychological well-being was modestly associated with depressive symptoms, education, gender, and age, while care-recipient cognitive status showed only weak links to anxiety. Education emerged as a consistent protective factor, whereas female gender and older age were associated with lower well-being. Although the Bonferroni correction eliminated significance in separate models, a complementary multivariate multiple regression confirmed global effects of education, caregiver gender, and depression across well-being domains. These findings emphasize the need for systematic psychological support for caregivers and call for larger, longitudinal studies to clarify causal mechanisms and additional protective factors.

## 1. Introduction

The aging of the population, a global demographic reality, has led to concerns about rising dementia prevalence. A 2019 study estimated that the number of people with dementia could reach 152.8 million by 2050 [1], although later work cautions that estimates vary depending on methodology and representativity [2,3,4]. According to the World Health Organization (WHO), 57 million people were diagnosed with dementia in 2021, with approximately 10 million new cases annually [5]. Prevalence doubles every five years after age 65 [6], being especially high in aging societies such as Japan, Italy, and Germany [7]. Alzheimer’s disease is the most common neurodegenerative disorder worldwide, followed by Parkinson’s disease [6,8,9], and the WHO reports rank dementia among the top three causes of death in Great Britain between 2021 and 2023 [10].

Dementia involves progressive neurodegeneration with hippocampal and cortical atrophy, neuronal and synaptic loss, and abnormal accumulations of amyloid-β and tau proteins, leading to severe cognitive decline, behavioral changes, and personality alterations [11,12,13,14]. Its most prominent symptoms include impairments in activities of daily living, behavioral problems, cognitive decline, and communication difficulties, making dementia a leading cause of disability and mortality among neurological conditions [15].

Caring for people with dementia imposes substantial emotional and physical demands. Surveys across Europe show that many caregivers provide extensive daily care (over 10 h in one-third of cases) and often receive inadequate information at diagnosis [16,17]. Many report worries about the future, sadness, depression, and loneliness, while meta-analyses confirm significantly higher stress, depression, and physical health problems among dementia caregivers compared with non-caregivers [18,19,20]. Caregiver well-being is shaped by both their own mental health and the condition of the care recipient, as well as by sociodemographic, relational, and individual factors such as coping, resilience, and pre-existing health [18,21,22,23,24,25].

Psychological well-being can be assessed across dimensions such as autonomy, self-acceptance, personal growth, and purpose in life [26,27,28]. Lower well-being has been linked to depression and anxiety [29], whereas adaptive coping strategies (e.g., social support, personal control) are associated with better outcomes [30]. Understanding how well-being, coping, and patient factors interact is critical for developing interventions [31]. Depression and anxiety remain the most common mental health problems among dementia caregivers [32], fueled by caregiving burden, limited support, and social isolation [33], and are further heightened by uncertainty, financial strain, and the progressive course of dementia [17,19].

### Rationale for the Current Study

Romania, like other Eastern European countries, faces rapid demographic aging but remains underrepresented in international caregiver research. Few studies have examined how Romanian caregivers’ psychological well-being is shaped by depressive and anxiety symptoms, socio-demographic factors, and the cognitive decline of care recipients. Addressing this gap is essential for developing context-appropriate interventions and public health strategies.

The present study, therefore, assessed the psychological well-being of family caregivers of individuals with dementia and investigated its associations with key factors, including depressive and anxiety symptoms, care-recipient cognitive function, and socio-demographic characteristics such as education, age, and gender. By clarifying these associations, this study contributes to the growing evidence base on the mental health of informal caregivers and highlights priorities for targeted support interventions in Romania.

## 2. Objectives

The objectives of this study were to assess the psychological well-being of family caregivers of individuals with dementia, to investigate its associations with depressive and anxiety symptoms and the cognitive functioning of care recipients, and to examine the contribution of socio-demographic characteristics such as education, age, and gender to caregiver well-being.

## 3. Materials and Methods

### 3.1. Study Design

This was a single-center, observational, analytical cross-sectional study. The study included 73 family caregivers of patients diagnosed with dementia who had accessed medical care for their care recipients between November 2023 and April 2024 in the Neurology-Psychiatry Department of the C.F.2 Clinical Hospital in Bucharest, Romania. Caregivers qualified for inclusion if they were 30 years of age or older, a family member of the patient, had provided caregiving for a minimum of six months, were caring for patients with dementia, and were capable of comprehending and completing the study instruments. Out of the 120 individuals who sought this service in the 6-month period, 73 subjects were selected based on the inclusion/exclusion criteria described above. Also, subjects who were providing care to a dementia patient but were not family members, as well as those who did not fully complete the questionnaire items, were excluded from the study.

### 3.2. Measures

Data collection instruments included Ryff’s Psychological Well-Being Scale, the PHQ-9, the COVI Scale, the Mini-Mental State Examination (MMSE, obtained from medical records), and an anthropological questionnaire (AQ) designed for this study to capture socio-demographic data.

The Romanian adaptation of Ryff’s Psychological Well-Being Scales (54 items) was used [26,27,34]. The scale assesses six dimensions—autonomy, personal growth, positive relations, self-acceptance, purpose in life, and environmental mastery—each measured by nine items. Responses are rated on a 6-point Likert scale (from strongly disagree to strongly agree), with 28 items reverse-coded. Subscale scores can be analyzed separately or combined into an overall index. The Ryff scale has been validated cross-culturally, including in Romania, and shows good reliability and factorial validity [28,35,36,37,38].

Caregivers’ depressive symptoms were measured using the Patient Health Questionnaire-9 (PHQ-9), where scores of 5–9 indicate mild, 10–14 moderate, 15–19 moderately severe, and 20–27 severe depression. The PHQ-9 is based on DSM-IV/DSM-5 diagnostic criteria and has demonstrated good reliability and validity in both clinical and research settings [39,40,41,42,43]. In Romania, it has been validated in multiple studies, showing usefulness for depression screening across different patient groups [44].

Anxiety was assessed using the COVI Scale, a clinician-rated instrument designed to evaluate the severity of anxiety symptoms. Scores are classified as follows: minimal or no anxiety (3–5), mild anxiety (6–8), moderate anxiety (9–11), and severe anxiety (12–15). The scale is widely used in both clinical and research contexts to quantify anxiety severity and to guide the development of targeted interventions for anxiety reduction [45].

Cognitive functioning of care recipients was evaluated using the Mini-Mental State Examination (MMSE), a widely recognized instrument for screening cognitive impairment. The MMSE includes 30 items covering orientation, attention, memory, and language, with total scores ranging from 0 to 30; lower scores indicate greater cognitive dysfunction, and a threshold of 24 or below is commonly used to suggest possible cognitive impairment. For this study, MMSE scores were extracted directly from patients’ medical records with institutional ethics approval; separate patient consent was not required as only de-identified data were used. The MMSE has been extensively validated across populations and is considered a standard in cognitive assessment [46,47,48].

Surveys (Ryff, PHQ-9, AQ) were self-administered in the physician’s office at C.F.2 Clinical Hospital, while the COVI was rated by the attending psychiatrist. Data collection took approximately 45 min per participant. A pilot test with 15 formal caregivers was conducted to refine the AQ; feedback was incorporated into the final version.

### 3.3. Procedure and Ethical Considerations

All participants provided written informed consent prior to study enrollment. Only caregivers signed consent forms, as MMSE scores for care recipients were obtained from clinical records. Access to these medical records was approved by the institutional ethics committee, and separate patient consent was not required because the data was de-identified. Data collection started after the study protocol was approved by the Ethics Committee of C.F.2 Clinical Hospital (Ref. Number: 1781/06.02.2023).

### 3.4. Statistical Analysis

Data were processed using IBM SPSS Statistics, version 13 (IBM Corp., Armonk, NY, USA), and Microsoft 365 Suite was used for tables and graphic design (Microsoft Corporation, Redmond, WA, USA). Descriptive statistics were used to summarize sample characteristics and scale scores. The distribution of continuous variables was examined using the Shapiro–Wilk and Kolmogorov–Smirnov tests. Bivariate associations were explored with Spearman correlations. To identify associated factors of caregiver psychological well-being, separate linear regression analyses were conducted for each Ryff subscale as the dependent variable. Independent variables included depressive symptoms (PHQ-9), anxiety (COVI), cognitive function (MMSE), and socio-demographic factors (age, gender, education, income, residence, care-recipient gender). Model fit was evaluated with R^2^, adjusted R^2^, and F tests. Variance inflation factors (VIF) were calculated to assess multicollinearity. A Bonferroni correction was applied across regression models to adjust for multiple testing. Statistical significance was set at *p* < 0.05.

## 4. Results

### 4.1. Descriptive Statistics

The study sample consisted of 73 family caregivers with a mean age of 57.1 years (SD = 10.4). Most participants were women (75.3%, *n* = 55), with men representing 24.7% (*n* = 18). The majority resided in urban areas (67.1%) and had at least completed high school (80.8%). Nearly four out of five families reported a monthly household income below 1000 EUR (78.1%), while 9.6% did not disclose income. Socio-demographic characteristics are presented in Table 1.

Descriptive analyses revealed that caregivers reported moderate to high levels of psychological well-being. Among the Ryff subscales, the highest mean score was observed for Environmental Mastery (M = 38.01, SD = 8.70), followed by Self-Acceptance (M = 37.78, SD = 8.16) and Positive Relations (M = 37.48, SD = 7.73). The lowest mean was recorded for Purpose in Life (M = 33.14, SD = 6.72), closely followed by Personal Growth (M = 33.42, SD = 8.68) (Figure 1).

The depression scores (PHQ-9) averaged 18.62 (SD = 6.30), indicating moderate levels of depressive symptoms. Anxiety scores were also elevated (COVI: M = 12.14, SD = 2.23). Care recipients demonstrated marked cognitive impairment, as reflected in the MMSE total score (M = 11.47, SD = 7.00).

Tests of normality (Shapiro–Wilk and Kolmogorov–Smirnov) indicated that several variables deviated from a normal distribution, including Purpose in Life, MMSE, PHQ-9, and COVI scores. Accordingly, Spearman’s rho correlations were used for these variables, while Pearson’s r was applied when normality assumptions were met.

### 4.2. Correlations Among Psychological Well-Being Dimensions (Ryff Scale)

Pearson correlation analysis demonstrated that all six Ryff dimensions were significantly and positively correlated (*p* < 0.001), supporting the internal coherence of the psychological well-being construct. The strongest associations were observed between Environmental Mastery and Self-Acceptance (r = 0.823) and Environmental Mastery and Positive Relationships (r = 0.824).

Personal Growth was also highly correlated with Positive Relationships (r = 0.793), and Purpose in Life showed strong positive relationships with all other well-being dimensions, particularly Self-Acceptance (r = 0.636) and Positive Relationships (r = 0.634).

Autonomy had moderate positive correlations with other Ryff dimensions and was positively associated with the caregiver’s educational level (r = 0.266, *p* = 0.023). No other significant associations were found between psychological well-being scores and income level, patient gender, or caregiver gender.

### 4.3. Correlations Between Cognitive Function, Depression, and Anxiety

Spearman correlation analysis revealed a significant negative correlation between patients’ MMSE score and caregiver anxiety level (MMSE–COVI: ρ = −0.273, *p* = 0.020), indicating that caregivers of patients with more severe cognitive decline experienced higher anxiety.

No significant associations were found between MMSE and caregiver depressive symptoms (ρ = −0.005, *p* = 0.968). The correlation between depressive and anxiety symptoms was positive but did not reach statistical significance (PHQ-9–COVI: ρ = 0.214, *p* = 0.069).

No significant correlations were observed between depression or anxiety scores and socio-demographic variables such as caregiver gender, education, income, or patient gender.

### 4.4. Regression Analysis

Separate linear regression models were conducted for each Ryff subscale, with caregiver depressive symptoms (PHQ-9), anxiety (COVI), patient cognitive status (MMSE), and socio-demographic factors (age, gender, education, income, residence, and patient gender) as independent variables (IV). The regression model fit for Ryff subscales is synthesized in Table 2.

For Self-Acceptance, the model accounted for 24.6% of the variance (R^2^ = 0.246, adj. R^2^ = 0.138). Higher depressive symptoms were associated with greater scores (β = 0.39, *p* = 0.009, *p*(Bonf) ≈ 0.508), whereas older age (β = −0.20, *p* = 0.028, *p*(Bonf) = 1.000) and being female (β = −4.88, *p* = 0.023, *p*(Bonf) = 1.000) were linked to lower scores.

Positive Relations were moderately associated with the IV, with the model explaining 23.1% of the variance (R^2^ = 0.231, adj. R^2^ = 0.121). Lower scores were associated with older age (β = −0.18, *p* = 0.042, *p*(Bonf) = 1.000) and female gender (β = −5.33, *p* = 0.011, *p*(Bonf) ≈ 0.619), while higher education showed a positive association (β = 3.64, *p* = 0.018, *p*(Bonf) ≈ 0.972).

For Autonomy, the model explained 22.3% of the variance (R^2^ = 0.223, adj. R^2^ = 0.112, omnibus *p* = 0.053). Higher caregiver education (β = 3.46, *p* = 0.008, *p*(Bonf) ≈ 0.423) and depressive symptoms (β = 0.32, *p* = 0.009, *p*(Bonf) ≈ 0.502) were positively associated with autonomy. Anxiety showed a negative trend, but this did not reach significance (*p* > 0.05).

The model for Personal Growth explained 21.2% of the variance (R^2^ = 0.212, adj. R^2^ = 0.099, omnibus *p* = 0.072). Older age was associated with lower scores (β = −0.21, *p* = 0.009, *p*(Bonf) ≈ 0.494), while higher education showed a positive relationship (β = 2.87, *p* = 0.033, *p*(Bonf) = 1.000).

In Environmental Mastery, the variables entered in the models accounted for 17.7% of the variance (R^2^ = 0.177, adj. R^2^ = 0.059). Lower scores were associated with female gender (β = −5.31, *p* = 0.032, *p*(Bonf) = 1.000), while higher education showed a borderline positive effect (β = 3.54, *p* = 0.050, *p*(Bonf) = 1.000).

Finally, Purpose in Life was not meaningfully related to the associated factors, with the model explaining only 8.1% of the variance (R^2^ = 0.081, adj. R^2^ = −0.050). No significant associations were detected (all *p* > 0.05, *p*(Bonf) = 1.000).

### 4.5. Model Refinement and Collinearity Analysis

Initial models included additional socio-demographic variables, such as household income category, patient sleeping arrangements, and patient gender. These were subsequently eliminated due to non-significance and redundancy. Their removal improved model parsimony without compromising explanatory power. After Bonferroni correction, none of the associations remained statistically significant, underscoring the restrictiveness of this method.

Collinearity diagnostics indicated no concerns, with VIF values ranging from 1.05 to 1.34 and corresponding tolerance values generally above 0.70. These results are well within accepted thresholds, confirming the absence of multicollinearity. To further minimize overlap, conceptually redundant associated factors (e.g., continuous versus categorical measures of depression) were tested in separate models, ensuring the robustness of coefficient estimates.

### 4.6. Multivariate Analyses

A multivariate multiple regression (MANOVA) was performed with the six Ryff well-being dimensions (autonomy, personal growth, positive relations, self-acceptance, purpose in life, environmental mastery) as dependent variables and depressive symptoms (PHQ-9), anxiety severity (COVI), care-recipient cognitive function (MMSE), caregiver age, gender, education, income, residence, and patient gender as independent variables.

The overall multivariate tests (Wilks’ λ) indicated that education (Wilks’ λ = 0.660, F(12,112) = 2.15, *p* = 0.0188), caregiver gender (Wilks’ λ = 0.789, F(6,56) = 2.49, *p* = 0.0330), and depressive symptoms (PHQ-9) (Wilks’ λ = 0.799, F(6,56) = 2.35, *p* = 0.0432) were significantly associated with the combined well-being profile. Caregiver age showed a trend-level effect (Wilks’ λ = 0.834, F(6,56) = 1.86, *p* = 0.1032). Anxiety (COVI), MMSE scores, income, residence, and patient gender were not significant (all *p* > 0.20).

Follow-up univariate analysis (Table 3) provided additional detail. Education was positively related to autonomy, personal growth, positive relations, and environmental mastery. Female caregivers scored lower on self-acceptance and positive relations. Older caregivers had lower scores on personal growth. Depressive symptoms remained positively associated with autonomy (β = 0.32, *p* = 0.009) and self-acceptance (β = 0.39, *p* = 0.009). In contrast, anxiety and MMSE showed no robust associations with any individual subscale after accounting for other IVs. The R^2^ values of the univariate models ranged from 0.08 to 0.25, reflecting modest explanatory power.

These findings strengthen the evidence that caregiver education, gender, and depression are consistently related to psychological well-being across multiple domains. While separate regression models with Bonferroni adjustment rendered all associations non-significant, the multivariate approach demonstrates that Bonferroni is overly conservative in this context and obscures meaningful patterns.

## 5. Discussions

This study examined the psychological well-being, depression, and anxiety levels of caregivers of patients with dementia and how these outcomes relate to socio-demographic factors and the cognitive status of the care recipients.

The strong intercorrelations among the Ryff psychological well-being dimensions are consistent with the previous literature emphasizing the multidimensional nature of well-being [26,28]. The particularly high associations between Environmental Mastery, Self-Acceptance, and Positive Relationships suggest that caregivers with a strong sense of environmental mastery and self-acceptance are more likely to maintain supportive social relationships, which serve as a critical resource for managing the ongoing stress associated with long-term caregiving [18,49].

Autonomy showed a moderate correlation with other well-being dimensions and a weaker correlation with caregiver education. The association between higher educational levels and greater caregiver autonomy suggests that educational attainment may enhance a sense of agency and control—factors recognized for their protective role against caregiver burnout [50,51,52]. No other statistically significant correlations were found between well-being scores and income, the gender of the patient, or the caregiver’s gender.

Earlier research has emphasized that caregiver burden increases with dementia severity, largely due to impairments in basic and instrumental activities of daily living (ADLs) [53,54,55]. Our findings are aligned, indicating that caregivers’ psychological well-being is more closely linked to their internal resources and the cognitive condition of the person they care for than to external socio-demographic factors. This underscores the importance of interventions that strengthen self-efficacy, emotional regulation, and coping skills among caregivers.

Regression analyses provided further insight, even though the models explained only modest proportions of variance (R^2^ ranging from 0.081 to 0.246). Consistent patterns emerged: higher education was positively associated with Autonomy, Personal Growth, Positive Relations, and Environmental Mastery; older age was linked to lower Personal Growth, Self-Acceptance, and Positive Relations; female caregivers reported lower levels of Self-Acceptance, Positive Relations, and Environmental Mastery. By contrast, anxiety and care-recipient cognitive status (MMSE) did not emerge as significant associated factors in the regression models, although bivariate analysis showed that lower MMSE scores were significantly related to higher caregiver anxiety—consistent with prior studies indicating that progressive cognitive decline intensifies caregiving demands and uncertainty [23,25,56,57,58]. In addition, depressive symptoms were associated with both Autonomy and Self-Acceptance, suggesting that mood disturbances influence key aspects of caregivers’ self-perceptions, findings in line with other studies [59,60].

Although several associations reached conventional significance (*p* < 0.05), none remained significant after Bonferroni correction. This highly conservative adjustment increases type II error when multiple outcomes are tested and may have masked meaningful associations. Thus, the results should be interpreted as indicative rather than definitive, but the consistency of the observed patterns provides important directions for intervention.

A visual synthesis of the tested associations is provided in Figure 2, which illustrates both significant and non-significant pathways between independent variables and the six Ryff dimensions. The figure highlights the central role of education, age, gender, and depression in shaping caregiver well-being, while also demonstrating the absence of effects for variables such as income, residence, and patient gender. Significant associations are reported in Table 4, complementing the graphical overview.

In our regression models, higher depressive symptoms were unexpectedly associated with higher scores on autonomy and self-acceptance. This contrasts with much of the existing literature, which typically links depression to diminished self-perception and agency [18,19]. One possible explanation is that some caregivers experiencing depressive symptoms may simultaneously engage in cognitive reappraisal or compensatory coping strategies that reinforce a sense of autonomy and self-reflection. Alternatively, the association may reflect shared method variance, as both constructs were assessed through self-report instruments, potentially inflating correlations. These findings should therefore be interpreted with caution and replicated in larger samples, ideally complemented by clinician-rated assessments of well-being.

Overall, the detrimental influence of depression across well-being domains, the positive role of education, and the vulnerability associated with female gender and older age highlight the dual need for systematic mental health screening and tailored support for family caregivers. Meanwhile, the domain-specific influence of anxiety, particularly its link with Purpose in Life at the trend level and with MMSE at the bivariate level, suggests that anxiety may selectively impair caregivers’ sense of meaning and control. These findings stress the importance of resilience-building strategies and targeted psychological support, particularly for those caring for patients with advanced cognitive impairment.

To address concerns regarding multiple testing, we complemented the separate regression models with a multivariate multiple regression (MMR). This approach tested the combined influence of correlates across all six Ryff dimensions simultaneously, thereby reducing the risk of inflated type II error introduced by Bonferroni correction. The MMR confirmed significant overall effects for caregiver education, caregiver gender, and depressive symptoms, even after accounting for the intercorrelated nature of the well-being subscales. These results strengthen the interpretation that education acts as a consistent protective factor, whereas female gender and depressive symptoms represent vulnerability markers for caregiver well-being. Thus, although Bonferroni adjustment masked significance in univariate models, the multivariate perspective supports the robustness of the observed patterns and helps mitigate the over-conservativeness of Bonferroni correction.

## 6. Conclusions

The psychological well-being of family caregivers of people with dementia was modestly associated with their own depressive symptoms, age, gender, and educational level, while the cognitive status of care recipients was only weakly related to caregiver anxiety. Education consistently emerged as a protective factor across several well-being dimensions, whereas female caregivers and older age were linked to lower scores in multiple domains. Although depressive symptoms were associated with self-acceptance and autonomy, the direction of this relationship warrants further investigation.

Overall, the separate regression models explained modest proportions of variance, and none of the associations remained significant after Bonferroni correction. This correction, while statistically rigorous, is highly conservative and likely obscured potentially meaningful relationships. To address this, we complemented the analysis with a multivariate multiple regression (MMR), which confirmed significant global effects of education, caregiver gender, and depression across the six well-being dimensions. These findings support the robustness of the observed patterns and underline their clinical relevance.

The results highlight the need for larger, longitudinal studies to clarify causal mechanisms and to identify additional factors—such as coping strategies, social support, and resilience—that may shape caregiver well-being. Nevertheless, the study contributes novel data from an under-researched context and emphasizes the importance of systematic psychological support for family caregivers in Romania.

## 7. Study Limits

Several limitations should be acknowledged when interpreting these findings. First, the cross-sectional design does not allow causal inference, and associations should be interpreted as correlational. For example, while higher education was associated with greater autonomy and personal growth, we cannot conclude whether education actively enhances well-being or whether individuals with greater well-being are more likely to pursue education. Similarly, the observed link between lower MMSE scores and higher caregiver anxiety at the bivariate level may reflect a bidirectional process that longitudinal studies could clarify.

Second, the sample size was modest (N = 73) and drawn from a single urban hospital in Bucharest, with most participants having medium or higher education. This composition limits the generalizability of the results to caregivers in rural or less advantaged settings.

Third, results were obtained mostly with self-report instruments (Ryff Scale, PHQ-9), which are susceptible to memory errors and social desirability bias. Anxiety was measured with the COVI scale through psychiatrist ratings, which introduces another layer of subjectivity. The unexpected positive association between depressive symptoms and self-acceptance, for example, may reflect such measurement artifacts or cultural patterns in reporting rather than a true effect, an unexpected pattern that requires cautious interpretation and further study.

From an analytical perspective, the models generally explained only modest proportions of variance (R^2^ between 0.08 and 0.25), suggesting that well-being is influenced by a broader range of variables not included here, such as coping strategies, social support, personality traits, or physical health. Moreover, the use of a Bonferroni correction—while conservative and methodologically sound—increased the risk of type II error, meaning that potentially meaningful associations (e.g., education and gender effects) did not retain statistical significance. Future studies may benefit from alternative methods, such as false discovery rate adjustments, which balance type I and type II errors more effectively.

Future research should address these limitations by employing larger, multi-center, and longitudinal designs to capture the dynamics of caregiver well-being over time and across diverse contexts. Expanding the models to include additional psychological and social variables and testing potential interaction effects (e.g., gender roles, caregiving intensity) could provide a richer and more nuanced understanding of caregiver well-being. Such work will be essential for developing tailored psychological and social interventions to support family caregivers of individuals with dementia.

## Figures and Tables

**Figure 1 healthcare-13-02235-f001:**
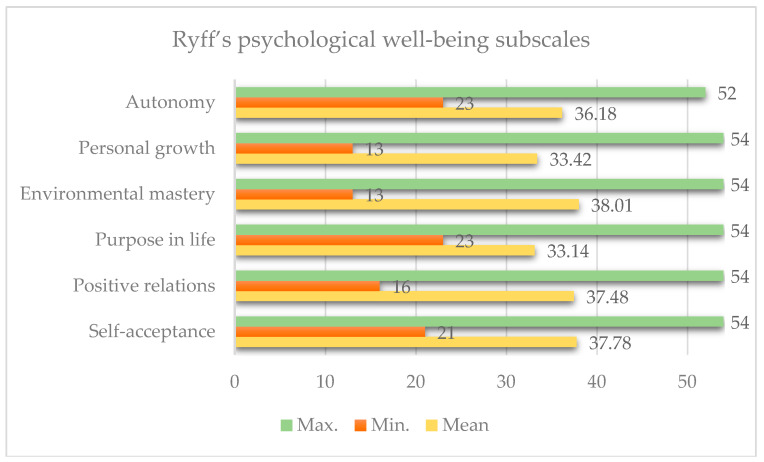
Descriptive statistics of Ryff’s dimensions.

**Figure 2 healthcare-13-02235-f002:**
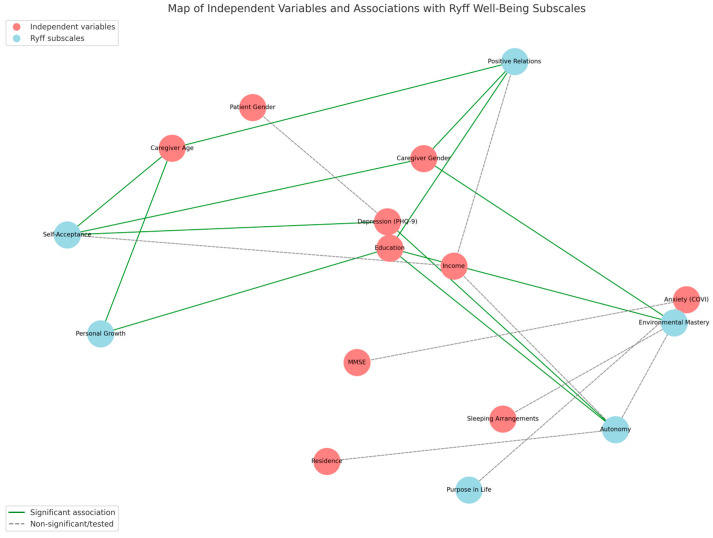
Conceptual map of independent variables and their associations with the six Ryff well-being dimensions.

**Table 1 healthcare-13-02235-t001:** Socio-demographical data.

Caregiver’s Characteristics	Variable	N (%)
Gender	Male	18 (24.7%)
	Female	55 (75.3%)
Origin	Rural	24 (32.9%)
	Urban	49 (67.1%)
Age (years)	Mean (SD)	57.12 (10.363)
	30–57	41 (56.2%)
	58–87	32 (43.8%)
Education	Secondary	14 (19.2%)
	High school	32 (43.8%)
	Higher education	27 (37%)
Declared family income (monthly)	Up to 400 EUR	14 (19.2)
	400–1000 EUR	43 (58.9%)
	Above 1000 EUR	9 (12.3%)
	Not declared	7 (9.6%)

**Table 2 healthcare-13-02235-t002:** Regression model fit for Ryff subscales.

Ryff Subscale	R^2^	Adj. R^2^
Self-Acceptance	0.246	0.138
Positive Relations	0.231	0.121
Autonomy	0.223	0.112
Personal Growth	0.212	0.099
Environmental Mastery	0.177	0.059
Purpose in Life	0.081	−0.050

**Table 3 healthcare-13-02235-t003:** Univariate follow-up regressions for Ryff well-being subscales.

Ryff Dimension	R^2^	Adj. R^2^	Education (*p*)	Gender (*p*)	Age (*p*)	Depression (*p*)	Anxiety (*p*)	MMSE (*p*)	Income (*p*)	Residence (*p*)	Patient Gender (*p*)
Autonomy	0.22	0.11	**0.008**	0.061	0.080	**0.009**	0.248	0.511	0.702	0.844	0.938
Personal Growth	0.21	0.10	**0.033**	0.149	**0.009**	0.091	0.211	0.497	0.822	0.902	0.993
Positive Relations	0.23	0.12	**0.018**	**0.011**	0.042	0.087	0.342	0.683	0.741	0.873	0.956
Self-Acceptance	0.25	0.14	0.119	**0.023**	0.028	**0.009**	0.414	0.776	0.850	0.921	0.933
Purpose in Life	0.08	−0.05	0.131	0.256	0.182	0.146	0.397	0.599	0.841	0.899	0.977
Environmental Mastery	0.18	0.06	**0.050**	**0.032**	0.074	0.054	0.305	0.642	0.812	0.914	0.986

Note. Values are *p*-values from univariate regression models (Type II ANOVA for categorical IV). Bold values indicate *p* < 0.05 (uncorrected).

**Table 4 healthcare-13-02235-t004:** Regression model significant associations for Ryff subscales.

Ryff Subscale	Significant Associations (*p* < 0.05)
Self-Acceptance	↑ Depression (β = 0.39, *p* = 0.009, *p*(Bonf) ≈ 0.508); ↓ Age (β = −0.20, *p* = 0.028, *p*(Bonf) = 1.000); ↓ Female (β = −4.88, *p* = 0.023, *p*(Bonf) = 1.000)
Positive Relations	↓ Age (β = −0.18, *p* = 0.042, *p*(Bonf) = 1.000); ↓ Female (β = −5.33, *p* = 0.011, *p*(Bonf) ≈ 0.619); ↑ Education (β = 3.64, *p* = 0.018, *p*(Bonf) ≈ 0.972)
Autonomy	↑ Education (β = 3.46, *p* = 0.008, *p*(Bonf) ≈ 0.423); ↑ Depression (β = 0.32, *p* = 0.009, *p*(Bonf) ≈ 0.502)
Personal Growth	↓ Age (β = −0.21, *p* = 0.009, *p*(Bonf) ≈ 0.494); ↑ Education (β = 2.87, *p* = 0.033, *p*(Bonf) = 1.000)
Environmental Mastery	↓ Female (β = −5.31, *p* = 0.032, *p*(Bonf) = 1.000); ↑ Education (β = 3.54, *p* = 0.050, *p*(Bonf) = 1.000)
Purpose in Life	None (all *p* > 0.05, *p*(Bonf) = 1.000)

## Data Availability

Data are available upon reasonable request.

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
