# Peer review of "Socio-Demographic Factors Linked to Psychological Well-Being in Dementia Caregivers"

_healthcare, 2025, doi:10.3390/healthcare13172235_

Round 1
Reviewer 1 Report
Comments and Suggestions for Authors
Correct some typographical and grammatical errors that were written as sticky notes. The introduction section should be shorter, and the discussion section should be more detailed.

Author Response
Our reply is attached. Thank you very much!

Reviewer 2 Report
Comments and Suggestions for Authors
The investigators have chosen a highly relevant topic. Since many people with dementia depend on caregivers, assessing caregiver well-being has relevance.
There are many ways that the reporting of this study should be enhanced.
The first sentence of the introduction requires a reference, in particular, the reference to the incidence of depression since it is not mentioned in the subsequent sentence.
Under Rationale for the study, provide a thorough rationale for study and remove reference to the sample size and measures. The rationale should then link to the next section, Objectives. After the objectives are stated, the research design chosen to address the objectives should be stated.
The Methods section needs to be organized.
Under study design, state the design and remove reference to data collection.
Clarify the inclusion criteria - note that 30 years of age is stated three times, but it looks like an error for the second occurrence.
Report the procedural methods to recruiting the caregivers. How were they identified, who engaged them in the consent process, and so forth.
Remove all references to the number of people who participated from all sections except the results including in the abstract.
Under measures, it appears some are self-reported. Was the anxiety of the caregiver assessed - why did a clinician assess the caregivers' anxiety. How were clinician assessments made, who made them, did the person with dementia provide informed consent.
Statistical analyses
Were demographic factors controlled in the regression? The results section should be rewritten after redoing the statistical analyses. It is reported that the assumption of normality was not met - how were the data and regression analyses adjusted to account for the lack of normality? Since only correlations and regression with a cross-sectional study were performed, the word, predict, cannot be used anywhere including the title. Only associations can be reported. The description of the analyses and the reporting of the results are unclear. It appears several regression analyses were performed? if so, consider Bronferronni correction
Results: The first paragraph of the results should include the dates of data collection, how many participants were approached (RR), how many declined and why, and how many participated, and the demographic description of the sample.
Please note that the "high levels of psychological well-being" (line 245) is a very remarkable finding since many caregivers often report low levels of well-being.
The results of the regression need to be reported, including the correlation matrix, beta, and so forth with details in tables and overview in the text.
Since the analyses are questionable, the discussion cannot be assessed. Based on the stated objective (line 144), the reader expects one multiple regression but the results reported do not align with this stated objective.
The limitations should include weaknesses associated with the study design.
The report should end with a conclusion.
Comments on the Quality of English LanguageSeveral words are not correct; for example sinaptic should be synaptic. The whole manuscript requires a careful review for appropriate words.
The authors did not define some abbreviations such as GB.
The subject-verb (eg This study aimed to..." does not make sense. The authors conducted the study for a particular purpose.
The word, data, is plural, though sometimes it is incorrectly used as if it were singular.
There are several single sentences that are not paragraphs.
Sentences should not end with a preposition (eg. The psychological well-being of caregivers is signifi-27cantly adjusted by their own mental health symptoms and the cognitive status of the individuals they care for."
Author Response
Our reply is attached. Thank you very much Reviewer 2 for the detailed and constructive feedback!

Reviewer 3 Report
Comments and Suggestions for Authors
After careful review, I have decided that the manuscript requires revisions. The study’s design and public health implications are valuable; however, several areas must be addressed to improve the clarity, methodological transparency, and overall presentation of your manuscript. Below are detailed comments to revisions:
Abstract
-
Correct spelling: "pacients" → "patients."
-
No mention of whether the tools used (Ryff’s Scale, PHQ-9, COVI Anxiety Scale) were validated or adapted.
Introduction
-
The introduction is too long and unfocused, containing redundant citations and ideas.
-
Since the study focuses on caregivers in Romania, the introduction should explain why this population is relevant or underrepresented in current literature.
-
Emphasize the specific gap this Romanian population study fills.
-
Although there are citations from 2024, some supporting data come from older studies (e.g., 2003, 2006). Consider balancing these with more recent data if available.
-
Correct spelling errors and remove redundant phrasing.
Methods
-
Clarify the study design by specifying the type of study (e.g., randomized controlled trial, exploratory, comparative study).
-
Include a sample size calculation.
Results
-
Improve clarity in descriptive statistics.
-
Clarify percentages in Table 1 (e.g., gender counts add up to 73, but percentages shown are 33% and 66%, totaling 99%). Consider including median and interquartile ranges for skewed variables.
-
Provide reference ranges or cutoffs for PHQ-9, COVI, and MMSE scores to contextualize mean values (e.g., indicate what score ranges correspond to moderate depression).
-
Clearly state the dependent variables for each regression model upfront.
-
Consistently indicate model fit statistics (e.g., R², adjusted R²) across all models.
Discussion
-
While interpretations of correlations and regression outcomes are logical, they could be expanded to include implications for caregiving practice or policy.
-
The Theoretical and Practical Implications section repeats content already discussed; consider merging it with the main discussion to improve flow and avoid redundancy.
-
Minor grammar and phrasing improvements are needed throughout.
Once the above revisions have been thoroughly addressed, we would be happy to reconsider your manuscript for publication.
Please include a point-by-point response letter outlining how each comment has been addressed and submit the revised manuscript via the journal’s submission portal.
We look forward to your resubmission.
Sincerely,
Comments on the Quality of English LanguageLanguage should be improved by an English professional.
Author Response
Our reply is attached. Thank you very much Reviewer 3 for your feedback!

Round 2
Reviewer 3 Report
Comments and Suggestions for Authors
Accept in present form.